# Defending Diffusion Models Against Membership Inference Attacks via Higher-Order Langevin Dynamics

## Abstract

Recent advances in generative artificial intelligence applications have raised new data security concerns. This paper focuses on defending diffusion models against membership inference attacks. This type of attack occurs when the attacker can determine if a certain data point was used to train the model. Although diffusion models are intrinsically more resistant to membership inference attacks than other generative models, they are still susceptible. The defense proposed here utilizes critically-damped higher-order Langevin dynamics, which introduces several auxiliary variables and a joint diffusion process along these variables. The idea is that the presence of auxiliary variables mixes external randomness that helps to corrupt sensitive input data earlier on in the diffusion process. This concept is theoretically investigated and validated on a toy dataset and the CIFAR-10 dataset using the Area Under the Receiver Operating Characteristic (AUROC) curves and the FID metric.

## 1 Introduction

Diffusion models (Sohl-Dickstein et al., 2015; Ho et al., 2020) have been shown to be fundamentally less susceptible to data security issues than other generative models such as GANs (Matsumoto et al., 2023). However, recent work has shown that they are still vulnerable to Backdoor Attacks, Membership Inference Attacks (MIA), and Adversarial Attacks (Truong et al., 2025). Defense against MIA is desirable, especially if the model is trained on sensitive data, such as medical data or sensitive Intellectual Property. The standard for privacy surrounding Diffusion Models is Differentially Private Diffusion Models (DPDM) (Dockhorn et al., 2023). DPDM combines the Differentially Private Stochastic Gradient Descent (DP-SGD) technique (Abadi et al., 2016) and continuous-time diffusion models (Song et al., 2020). The quality of the samples generated with this method is directly related to the level of privacy chosen. This is commonly known as the privacy versus utility tradeoff.

Concurrently, the use of critically-damped and higher-order Langevin dynamics has been explored in other works with regards to continuous-time diffusion models. The seminal work of Dockhorn et al. (2021) introduced critically-damped Langevin dynamics (CLD), where a single auxiliary variable denoted *velocity* was augmented to the diffusion process to smooth the stochastic process trajectories. Smoothing is often desirable because it resembles the continuity of real-world data. Third-Order Langevin-Dynamics (TOLD) were introduced (Shi & Liu, 2024a) after CLD to add another auxiliary variable called *acceleration*, which had the effect of adding an extra smoothing component to the data. The authors additionally proposed in theory how to implement Higher-Order Langevin dynamics (HOLD) (Shi & Liu, 2024a). Since its invention, TOLD has been used in audio generation (Shi & Liu, 2024b) and image restoration tasks (Shi & Liu, 2024c). One further improvement to TOLD/HOLD is to reparameterize so that the diffusion process is critically-damped, resulting in critically-damped higher-order Langevin dynamics (HOLD++) (Sterling & Bugallo, 2025). It can be shown that critical damping is an optimal strategy in terms of convergence of the forward process. This makes HOLD++ an ideal tool to analyze how model dimensionality, among other factors, influences Membership Privacy.

Around the same time differential privacy was applied to diffusion models, there have been independently-developed attacks targeting diffusion models. The first such MIA, to our knowledge, was SecMI (Duan et al., 2023). It was developed only to target discrete time diffusion models. It uses the diffusion model's trained score network to approximate the forward and backward processes as deterministic processes, and it exploits the fact that the score network is optimized only on training data. The same authors further refine SecMI to Proximal Initialization (PIA) (Kong et al., 2024). It exploits the same nature of the score network that SecMI does, but it also provides a continuous-time version.

The goal of this work is to enhance the defenses of diffusion models against membership inference attacks, beyond standard differential privacy. Currently, the main defenses against membership inference attacks fall into the categories of differential privacy, $L_2$ regularization, and knowledge distillation (Truong et al., 2025). Our focus is on applying Critically-Damped Higher-Order Langevin Dynamics (HOLD++) (Sterling et al., 2025) to achieve a base level of differential privacy, and arguing that it theoretically defends against real-world membership inference attacks. This theoretical defense is validated on both a toy and the CIFAR-10 dataset (Krizhevsky & Hinton, 2009).

## 2 Background

Here we will briefly review how traditional continuous diffusion models (Song et al., 2020) apply to PIA; PIA will be used as a representative of such membership inference attacks. Diffusion models are a method of generating samples from an unknown intractable data distribution. They possess a forward process that transforms training data into noise, for the purpose of learning the score, and a backward process, for the purpose of generating synthetic samples from the data distribution. If the forward process of our model is $d\mathbf{x}_t = \mathbf{f}_t(\mathbf{x}_t)dt + g_t d\mathbf{w}$, then the deterministic reverse process is $d\mathbf{x}_t = \left(\mathbf{f}_t(\mathbf{x}_t) - \frac{1}{2}g_t^2 \nabla_{\mathbf{x}_t} \log p_t(\mathbf{x})\right) dt$ (Song et al., 2020). In practice, $\nabla_{\mathbf{x}_t} \log p_t(\mathbf{x}_t)$ is estimated with a neural network $s_\theta(\mathbf{x}_t, t)$. The PIA approach is to calculate the following attack metric for different data points:

$$R_{t,p} = \left\| \mathbf{f}_t(\mathbf{x}_t) - \frac{1}{2}g_t^2 \mathbf{s}_\theta(\mathbf{x}_t, t) \right\|_p,$$

where $||.||_p$ denotes the $p$-norm. The metric may be interpreted as the jump-size of the process. Data points with a comparatively lower attack metric are more likely to be within the training dataset because $\mathbf{s}_\theta(\mathbf{x}_t, t)$ were trained with them. Therefore, PIA thresholds this metric and uses threshold testing to classify training and hold out data.

## 3 Problem Formulation

This section will review HOLD++ and how to apply PIA to this specific diffusion method. It is argued here that HOLD++ is better at defending against PIA than traditional diffusion models because of its structure. Following Sterling et al. (2025) and the previous section, we define the forward SDE of HOLD++ as:

$$d\mathbf{x}_t = \mathcal{F}\mathbf{x}_t dt + \mathcal{G}d\mathbf{w},$$

$$\mathbf{F} = \sum_{i=1}^{n-1} \gamma_i \left(\mathbf{E}_{i,i+1} - \mathbf{E}_{i+1,i}\right) - \xi\mathbf{E}_{n,n}, \quad \mathbf{G} = \sqrt{2\xi L^{-1}}\mathbf{E}_{n,n},$$

$$\mathcal{F} = \mathbf{F} \otimes \mathbf{I}_d, \quad \mathcal{G} = \mathbf{G} \otimes \mathbf{I}_d.$$

where $\mathbf{w}$ is a standard Brownian motion, $\mathbf{E}_{i,j}$ is the zero matrix with a single 1 position $(i,j)$, and $\gamma_1, \dots \gamma_{n-1}, \xi$ are HOLD++ parameters. For example when $n = 3$:

$$\mathbf{F} = \begin{bmatrix} 0 & \gamma_1 & 0 \\ -\gamma_1 & 0 & \gamma_2 \\ 0 & -\gamma_2 & -\xi \end{bmatrix}, \quad \mathbf{G} = \begin{bmatrix} 0 & 0 & 0 \\ 0 & 0 & 0 \\ 0 & 0 & \sqrt{2\xi L^{-1}} \end{bmatrix}.$$

Here, the data variable is represented by $\mathbf{q}_0$, auxiliary variables are drawn according to $\mathbf{p}_0, \mathbf{s}_0, \ldots \sim \mathcal{N}(\mathbf{0}, \beta L^{-1}\mathbf{I})$, and $\mathbf{x}_0 = (\mathbf{q}_0^T, \mathbf{p}_0^T, \mathbf{s}_0^T, \ldots)^T$. It is shown in detail in Sterling et al. (2025) that the mean and covariance of the forward process are given by:

$$\begin{aligned}\boldsymbol{\mu}_t &= \exp(\mathcal{F}t)\mathbf{x}_0, \\ \boldsymbol{\Sigma}_t &= L^{-1}\mathbf{I} + \exp(\mathcal{F}t)\left(\boldsymbol{\Sigma}_0 - L^{-1}\mathbf{I}\right)\exp(\mathcal{F}t)^T,\end{aligned} \tag{1}$$

where $\exp(\cdot)$ is the matrix exponential map. One may sample from this distribution by taking the Cholesky Decomposition of $\boldsymbol{\Sigma}_t$, $\mathbf{L}_t$, and performing

$$\mathbf{x}_t = \boldsymbol{\mu}_t + \mathbf{L}_t\boldsymbol{\epsilon}, \tag{2}$$

where $\boldsymbol{\epsilon} = (\boldsymbol{\epsilon}_1^T, \boldsymbol{\epsilon}_2^T, \ldots \boldsymbol{\epsilon}_n^T)^T$ and $\boldsymbol{\epsilon}_1, \boldsymbol{\epsilon}_2 \ldots \boldsymbol{\epsilon}_n \sim \mathcal{N}(\mathbf{0}, \mathbf{I})$. When the PIA attack metric $R_{t,p}$ is adapted to the HOLD++ SDE, it becomes:

$$R_{t,p} = \left\|\mathcal{F}\mathbf{x}_t - \frac{1}{2}\mathcal{G}\mathcal{G}^T\mathbf{S}_\theta(\mathbf{x}_t, t)\right\|_p,$$

where $\mathbf{S}_\theta(\mathbf{x}_t, t) = (\mathbf{0}^T, \ldots \mathbf{0}^T, \mathbf{s}_\theta(\mathbf{x}_t, t)^T)^T$. The true scores of the first $(n-1)d$ entries are replaced with "$\mathbf{0}$" because they all cancel with $\mathcal{G}\mathcal{G}^T$. In this expression $\mathbf{x}_t$ is estimated deterministically with equation 2 using $\mathbf{s}_\theta(\mathbf{x}_0, 0)$ to estimate $\boldsymbol{\epsilon}_n$ with $\boldsymbol{\epsilon}_n \approx -\mathbf{s}_\theta(\mathbf{x}_0, 0)\mathbf{L}_0[-1, -1]$, where $\mathbf{L}_0[-1, -1]$ denotes the matrix element in the final row and column. We may even further simplify the attack metric as: $R_{t,p} = \left\|\mathcal{F}\mathbf{x}_t - \xi L^{-1}\mathbf{S}_\theta(\mathbf{x}_t, t)\right\|_p$.

The specifics of this attack using HOLD++ are summarized in Algorithm 1. With regular diffusion processes, $\mathbf{s}_\theta(\mathbf{x}_0, 0)$ is all one would need to estimate $\mathbf{x}_t$, but in the HOLD++ context, this quantity only informs us of the score function of the last auxiliary variable. Ideally, one would use $\boldsymbol{\epsilon}_1, \boldsymbol{\epsilon}_2, \ldots \boldsymbol{\epsilon}_{n-1} \sim \mathcal{N}(\mathbf{0}, \mathbf{I}_d)$ to match the true distribution of $\mathbf{x}_t$, but doing so defeats the purpose of using a deterministic attack metric. Therefore, the best thing one can do is set $\boldsymbol{\epsilon}_1, \boldsymbol{\epsilon}_2, \ldots \boldsymbol{\epsilon}_{n-1} = \mathbf{0}$. This work has attempted both, but only presents the results for $\boldsymbol{\epsilon}_1, \boldsymbol{\epsilon}_2, \ldots \boldsymbol{\epsilon}_{n-1} = \mathbf{0}$ as they are a more effective attack. Additionally, this work sets the auxiliary variables to zero during attack time, to avoid additional randomness. The attack metric involving $\mathcal{G}\mathcal{G}^T$ derives from the reverse deterministic process of the forward SDE Song et al. (2020).

---

**Algorithm 1** PIA Attack with HOLD++

1: **Input:** Data point $\mathbf{q}_0$ and Score Network $\mathbf{s}_\theta$, threshold $\tau$.
2: $\mathbf{x}_0 \leftarrow (\mathbf{q}_0^T, \mathbf{0}^T, \mathbf{0}^T, \ldots \mathbf{0}^T)^T$
3: **for** $k = 1$ to $n_{time}$ **do**
4:     $t \leftarrow (k-1)T/n_{time}$, Calculate $\boldsymbol{\mu}_t, \boldsymbol{\Sigma}_t, \mathbf{L}_t$ using equation 1.
5:     $\boldsymbol{\epsilon}_n \leftarrow -\mathbf{s}_\theta(\mathbf{x}_0, 0)\mathbf{L}_0[-1, -1]$
6:     $\boldsymbol{\epsilon}_{full} \leftarrow \begin{pmatrix}\mathbf{0}^T, & \mathbf{0}^T, & \ldots & \mathbf{0}^T, & \boldsymbol{\epsilon}_n^T\end{pmatrix}^T$
7:     $\mathbf{x}_t \leftarrow \boldsymbol{\mu}_t + \mathbf{L}_t\boldsymbol{\epsilon}_{full}$
8:     $R[n_{time}] \leftarrow \left\|\mathcal{F}\mathbf{x}_t - \xi L^{-1}\mathbf{S}_\theta(\mathbf{x}_t, t)\right\|_p$
9: **end for**
10: Hypothesis Test used to generate ROC Curves:
11: $\bar{R} \leftarrow \frac{1}{n_{time}}\sum_{k=1}^{n_{time}} R[k]$, **is\_in\_training\_set** $\leftarrow \bar{R} < \tau$

---

## 4 Methodology

This section rigorously proves that HOLD++ is Rényi Differentially Private and that this bound only depends on $\epsilon_{\text{num}}$, a variance addition to the data that ensures numerical stability. The same modification works to achieve differential privacy on traditional continuous diffusion models, but at the end of the section we demonstrate that this differential privacy, coupled with HOLD++'s non-deterministic score function, helps further deter MIAs for HOLD++. The Rényi divergence between two probability distributions $P$ and $Q$ is defined as:

$$D_\alpha(P \| Q) = \frac{1}{\alpha - 1}\log \mathbb{E}_{\mathbf{y} \sim Q}\left[\left(\frac{P(\mathbf{y})}{Q(\mathbf{y})}\right)^\alpha\right].$$

Rényi-Differential-Privacy is defined for a random mechanism $f$ as follows.

**Definition 4.1** $f : \mathcal{D} \rightarrow \mathbb{R}$ *has* $(\alpha, \epsilon)$ *Rényi Differential Privacy, if for adjacent* $X, X' \in \mathcal{D}$*, it follows that:* $D_\alpha(f(X) \mid\mid f(X')) \leq \epsilon$.

In our case, the random mechanism is defined by $f(\mathbf{x}) = \exp(\mathcal{F}t)\mathbf{x} + \boldsymbol{\eta}$, where $\boldsymbol{\eta} \sim \mathcal{N}(\mathbf{0}, \boldsymbol{\Sigma}_t)$. To compute the Rényi Divergence applied to $f$, we consider the distribution $P$ of a random variable $\mathbf{y}$ outputted by $f$, and the distribution $Q$ of a random variable $\mathbf{y} + \mathbf{v}$ outputted by $f$, where $\mathbf{v}$ is the maximum difference between any two adjacent data points. Specifically,

$$P(\mathbf{y}) = \mathcal{N}(\mathbf{y} \mid \exp(\mathcal{F}t)\mathbf{x}, \boldsymbol{\Sigma}_t), \quad Q(\mathbf{y}) = \mathcal{N}(\mathbf{y} + \mathbf{v} \mid \exp(\mathcal{F}t)\mathbf{x}, \boldsymbol{\Sigma}_t),$$

$$\mathbf{v} \in \{\mathbb{R}^{n \times d} \mid \mathbf{v}^T \boldsymbol{\Sigma}_t^{-1} \mathbf{v} \leq \Delta f_t\}, \quad \text{and}$$

$$\Delta f_t = \max_{\mathbf{y}, \mathbf{z} \in \mathcal{D}} (\mathbf{y} - \mathbf{z})^T \exp(\mathcal{F}t)^T \boldsymbol{\Sigma}_t^{-1} \exp(\mathcal{F}t)(\mathbf{y} - \mathbf{z}).$$

The following theorem is adapted from Mironov (2017) in the non-isotropic Gaussian case:

**Lemma 4.1** *The Random Mechanism* $f(\mathbf{x}) = \exp(\mathcal{F}t)\mathbf{x} + \boldsymbol{\eta}$ *where* $\boldsymbol{\eta} \sim \mathcal{N}(\mathbf{0}, \boldsymbol{\Sigma}_t)$ *satisfies* $RDP(\alpha, \frac{\alpha \Delta f_t}{2})$.

**Proof.**

Start by computing:

$$\mathbb{E}_{\mathbf{y} \sim Q} \left( \frac{P_t(\mathbf{y})}{Q_t(\mathbf{y})} \right)^\alpha = \int_{\mathbf{y} \in \mathbb{R}^{nd}} \frac{P_t(\mathbf{y})^\alpha}{Q_t(\mathbf{y})^\alpha} Q_t(\mathbf{y}) d\mathbf{y}.$$

After making a change of variables $\mathbf{u} = \mathbf{y} - \exp(\mathcal{F}t)\mathbf{x}$ the above expression becomes:

$$(2\pi)^{-nd/2} \det(\boldsymbol{\Sigma}_t)^{-1/2} \int_{\mathbf{u} \in \mathbb{R}^{nd}} \exp \left( \left( \frac{\alpha - 1}{2} \right) (\mathbf{u} + \mathbf{v})^T \boldsymbol{\Sigma}_t^{-1} (\mathbf{u} + \mathbf{v}) - \frac{\alpha}{2} \mathbf{u}^T \boldsymbol{\Sigma}_t^{-1} \mathbf{u} \right) d\mathbf{u}.$$

Note the identity: $\mathbf{u}^T \boldsymbol{\Sigma}_t^{-1} \mathbf{u} - 2(\alpha - 1)\mathbf{v}^T \boldsymbol{\Sigma}_t^{-1} \mathbf{u} = (\mathbf{u} - (\alpha - 1)\mathbf{v})^T \boldsymbol{\Sigma}_t^{-1} (\mathbf{u} - (\alpha - 1)\mathbf{v}) - (\alpha - 1)^2 \mathbf{v}^T \boldsymbol{\Sigma}_t^{-1} \mathbf{v}$. This allows us to complete the square and evaluate the expectation:

$$D_\alpha(P_t \mid\mid Q_t) = \frac{1}{\alpha - 1} \log \exp \left( \frac{(\alpha - 1) + (\alpha - 1)^2}{2} \mathbf{v}^T \boldsymbol{\Sigma}_t^{-1} \mathbf{v} \right) = \frac{\alpha}{2} \mathbf{v}^T \boldsymbol{\Sigma}_t^{-1} \mathbf{v} \leq \frac{\alpha \Delta f_t}{2}.$$

$\square$

Now, define $\mathbf{R}_t = (\exp(\mathcal{F}t)^T \boldsymbol{\Sigma}_t^{-1} \exp(\mathcal{F}t))^{-1}$, the effective correlation matrix. Using the derived formula for $\boldsymbol{\Sigma}_t$ and some algebraic simplifications: $\mathbf{R}_t = L^{-1} \left( \exp(\mathcal{F}t)^T \exp(\mathcal{F}t) \right)^{-1} + \boldsymbol{\Sigma}_0 - L^{-1}\mathbf{I}$. Now:

$$\Delta f_t = \max_{\mathbf{y}, \mathbf{z} \in \mathcal{D}} (\mathbf{y} - \mathbf{z})^T \mathbf{R}_t^{-1} (\mathbf{y} - \mathbf{z}).$$

**Lemma 4.2** $\Delta f_t$ *monotonically decreases with* $t$.

**Proof.** $\mathcal{F}$ is negative definite, and the following formula for the derivative of an inverse matrix: $\frac{d\mathbf{R}_t^{-1}}{dt} = -\mathbf{R}_t^{-1} \frac{d\mathbf{R}_t}{dt} \mathbf{R}_t^{-1}$. One may use these two details to prove the following equation that implies that $\Delta f_t$ monotonically decreases with $t$ and the maximum $\Delta f_t$ occurs at $t = 0$.

$$\frac{d\Delta f_t}{dt} = \max_{\mathbf{y}, \mathbf{z} \in \mathcal{D}} (\mathbf{y} - \mathbf{z})^T \frac{d\mathbf{R}_t^{-1}}{dt} (\mathbf{y} - \mathbf{z}) < 0.$$

$\square$

It follows algebraically that $\mathbf{R}_0 = \boldsymbol{\Sigma}_0$, thus the maximum $\Delta f_t$ is $\max_{\mathbf{y}, \mathbf{z} \in \mathcal{D}} (\mathbf{y} - \mathbf{z})^T \boldsymbol{\Sigma}_0^{-1} (\mathbf{y} - \mathbf{z})$. $\boldsymbol{\Sigma}_0 = \mathbf{diag}(\epsilon_{\text{num}}, \beta L^{-1}, \ldots, \beta L^{-1})$, where $\epsilon_{\text{num}}$ is a small initial variance for the position component. In practice, $\epsilon_{\text{num}} \ll \beta L^{-1}$, therefore:

$$\Delta f_t \leq \Delta f_0 = \max_{\mathbf{y}, \mathbf{z} \in \mathcal{D}} (\mathbf{y} - \mathbf{z})^T \boldsymbol{\Sigma}_0^{-1} (\mathbf{y} - \mathbf{z})$$

$$\approx \max_{\mathbf{y}, \mathbf{z} \in \mathcal{D}} (\mathbf{y} - \mathbf{z})^T \text{diag}(1/\epsilon_{\text{num}}, 0, \ldots 0)(\mathbf{y} - \mathbf{z}) = \frac{\Delta_2 f}{\epsilon_{\text{num}}},$$

where $\Delta_2 f = \max_{\mathbf{y}, \mathbf{z} \in \mathcal{D}} ||\mathbf{y} - \mathbf{z}||^2$ is the regular data sensitivity (excluding the auxiliary variables) used in other works. We may derive an upper bound on the true privacy loss, the Rényi Divergence between marginals $P_t(\mathbf{q}_t)$ and $Q_t(\mathbf{q}_t)$:

$$D_\alpha(P_t(\mathbf{q}_t) \,||\, Q_t(\mathbf{q}_t)) \leq D_\alpha(P_t \,||\, Q_t) \leq \frac{\alpha \Delta f_t}{2} \leq \frac{\alpha \Delta f_0}{2} \approx \frac{\alpha \Delta_2 f}{2 \epsilon_{\text{num}}}$$

The first inequality is true because marginals admit lower Rényi Divergence than joint distributions. This implies that HOLD++ is Rényi differentially private for $\epsilon = \frac{\alpha \Delta f_0}{2}$. We note that this bound is overly conservative, as it also bounds the joint privacy loss including all the auxiliary variables $\mathbf{p}_0, \mathbf{s}_0 \ldots$ etc. In real world experiments when $\epsilon_{\text{num}} \ll \beta L^{-1}$, the privacy loss simplifies. This strongly resembles the bound derived in Mironov (2017). The obvious problem is that small $\epsilon$ and $\epsilon_{\text{num}}$ cannot be achieved at the same time. Therefore, instead of solely relying on Differential Privacy, we argue that presence of auxiliary variables helps prevent an attacker from inferring membership. Having proven that privacy loss is maximized at the beginning of the diffusion process, we consider the Mean Squared Error (MSE) between $\mathbf{x}_{guess} = (\mathbf{q}_0^T, \mathbf{0}^T)^T$ and $\mathbf{x}_{truth} = (\mathbf{q}_0^T, \beta L^{-1} \mathbf{z}^T)^T$, where $\mathbf{q}_0$ is a point in the training data set and $\mathbf{z} \sim \mathcal{N}(\mathbf{0}, \mathbf{I}_{n-1})$. It follows: $\mathbb{E}(||\mathbf{x}_{guess} - \mathbf{x}_{truth}||^2) = \beta L^{-1}(n-1)$. This implies that the MSE may be "tuned" by the forward diffusion process by adjusting $\beta$, $L^{-1}$, and $n$, trading off model complexity, sample quality, and privacy leakage.

## 5  Experiments and Results

The theoretical section claims that PIA can be defended against using higher model orders $n$ and higher starting variances $\beta L^{-1}$. This section seeks to validate this claim on the Swiss Roll and CIFAR-10 datasets. The validation metric that this paper primarily uses is the Area Under the ROC curve (AUROC) that comes from running PIA. An AUROC close to 1.0 indicates that the attack can perfectly differentiate training versus holdout data points, whereas an AUROC close to 0.5 indicates that the attack does not do better than randomly guessing. The code is publicly available in the supplementary material.

Regarding the Swiss Roll dataset, the training and validation datasets are taken to be non-overlapping. Independent sessions are run in Figure 1 for differing $n$, $\beta$, and $\epsilon_{\text{num}}$ with fixed $L^{-1} = 1$. These runs were repeated 25 times to obtain confidence intervals and performed with 40,000 training epochs. A fully connected feedforward neural network was used with ReLU activation, layer normalization, and a total depth of 15 layers. Please, see the supplementary material for full architectural details. As predicted, AUROC tends to decrease with increasing $n$ and $\beta$. Notably, for $\beta = 2, 10$, the AUROC 95% confidence intervals do not overlap, suggesting that as $\beta$ increases, the pairwise differences in AUROC grow more statistically significant as one changes the order of the model $n$. Figure 2 illustrates how privacy loss is distributed over diffusion time, further demonstrating that higher model orders $n$ are more resistant to MIA, with vulnerabilities more time-localized. All of these results demonstrate that the implicit trade-off one makes under this defense scheme is model order $n$, variance factor $\beta$, privacy leakage controlled by $\epsilon_{\text{num}}$, AUROC, and visible spiral quality.

The rest of this section describes the experiments run on the CIFAR-10 dataset. The Fréchet Inception Distance (FID) metric (Heusel et al., 2017) is used to evaluate sample quality. This work, to our knowledge, is the first that attempted running the continuous proximal initialization attack on any real-world image dataset, as the PIA manuscript used grad-TTS (Popov et al., 2021) and the LJ Speech dataset for their continuous attack. The PIA manuscript only used CIFAR-10 to validate the discrete time version of their attack. Our first finding is that the continuous PIA is ineffective on continuous diffusion models trained

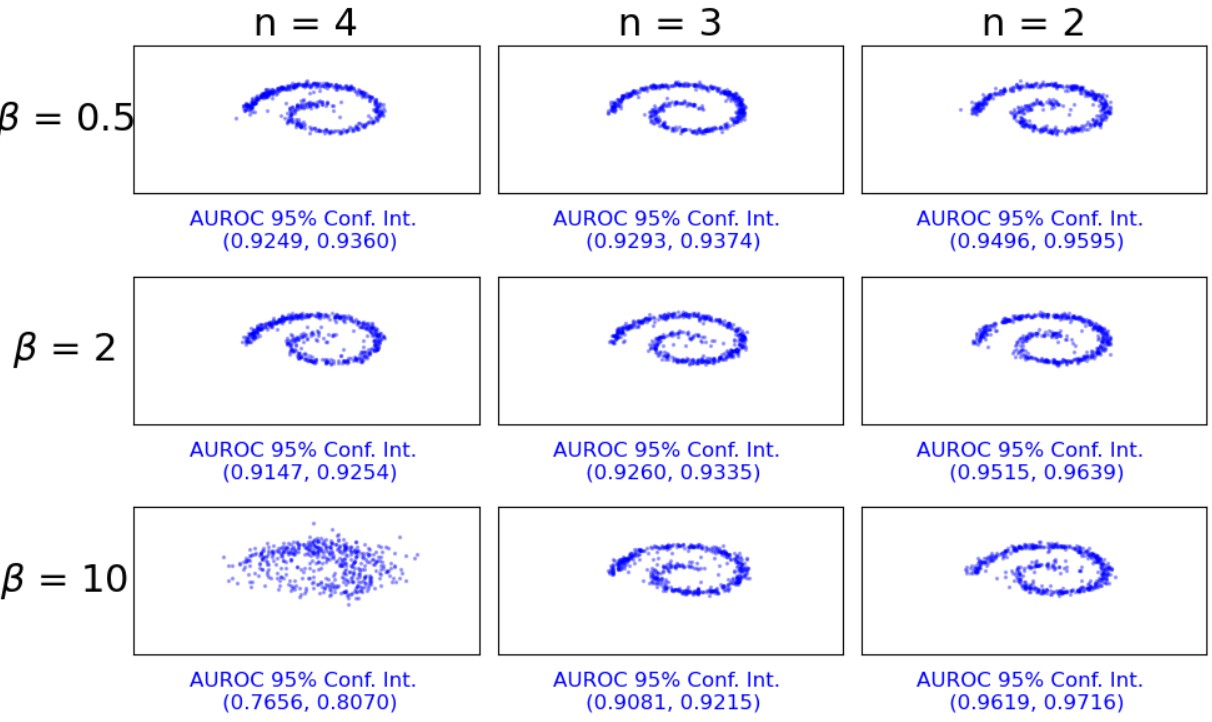

Figure 1: Generated Spirals grouped by model order $n$, variance factor $\beta$, and $\epsilon_{\mathrm{num}}$ for $L^{-1} = 1$. 95% confidence intervals of the AUROC's with 25 sample runs are presented.

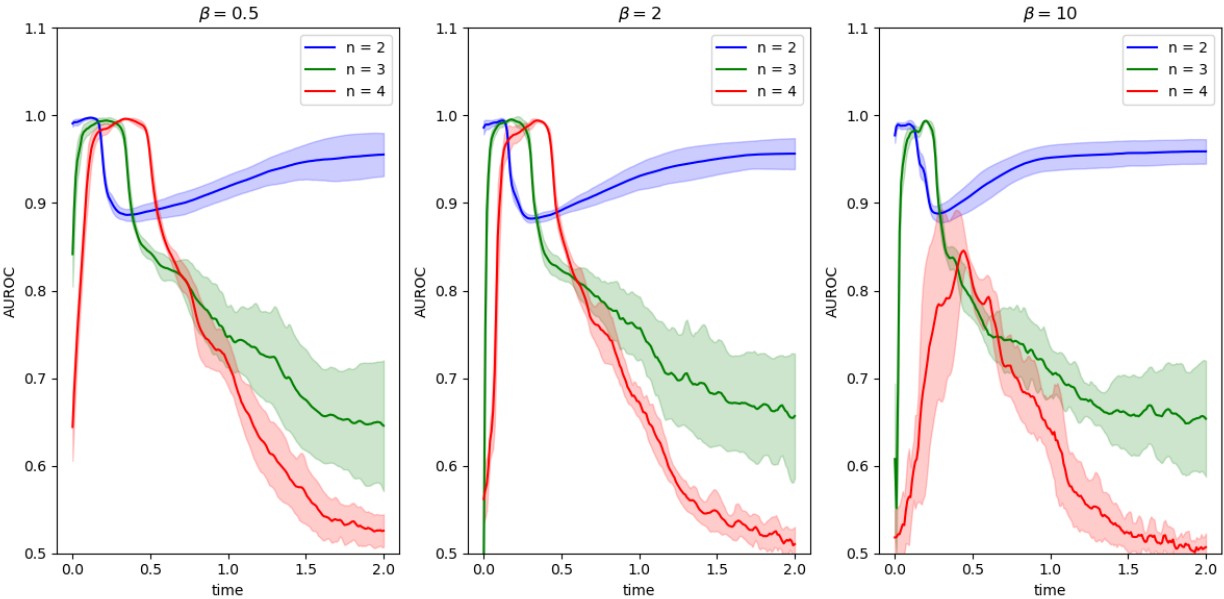

Figure 2: AUROC with 95% confidence intervals for $n$ as a function of diffusion time for spiral dataset. These are obtained by directly thresholding $R$ (not $\bar{R}$) referring to Algorithm 1.

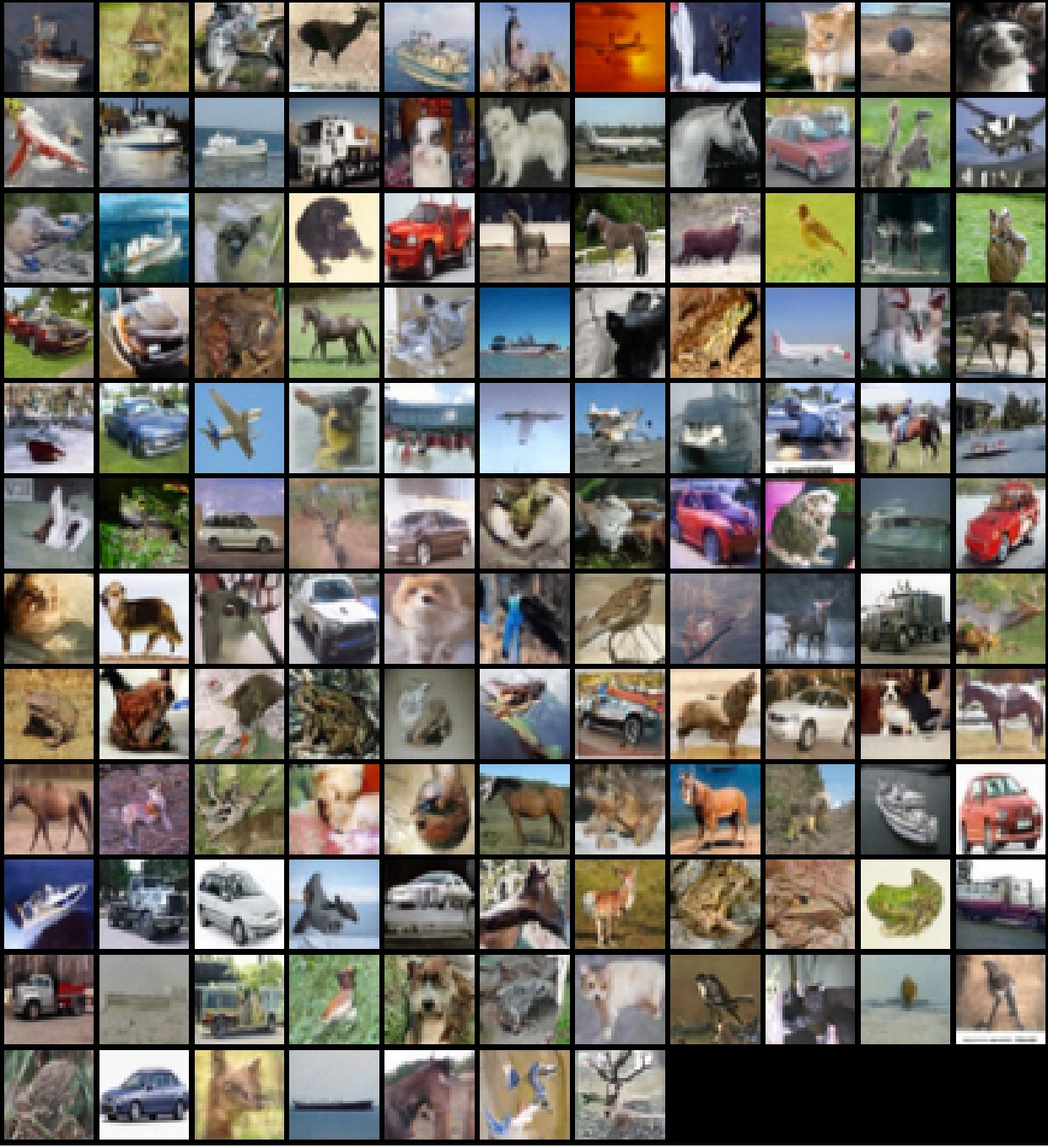

Figure 3: Generated CIFAR-10 samples from a VPSDE Diffusion Model with an FID of 7.03 and AUROC of 0.503 under the continuous proximal initialization attack.

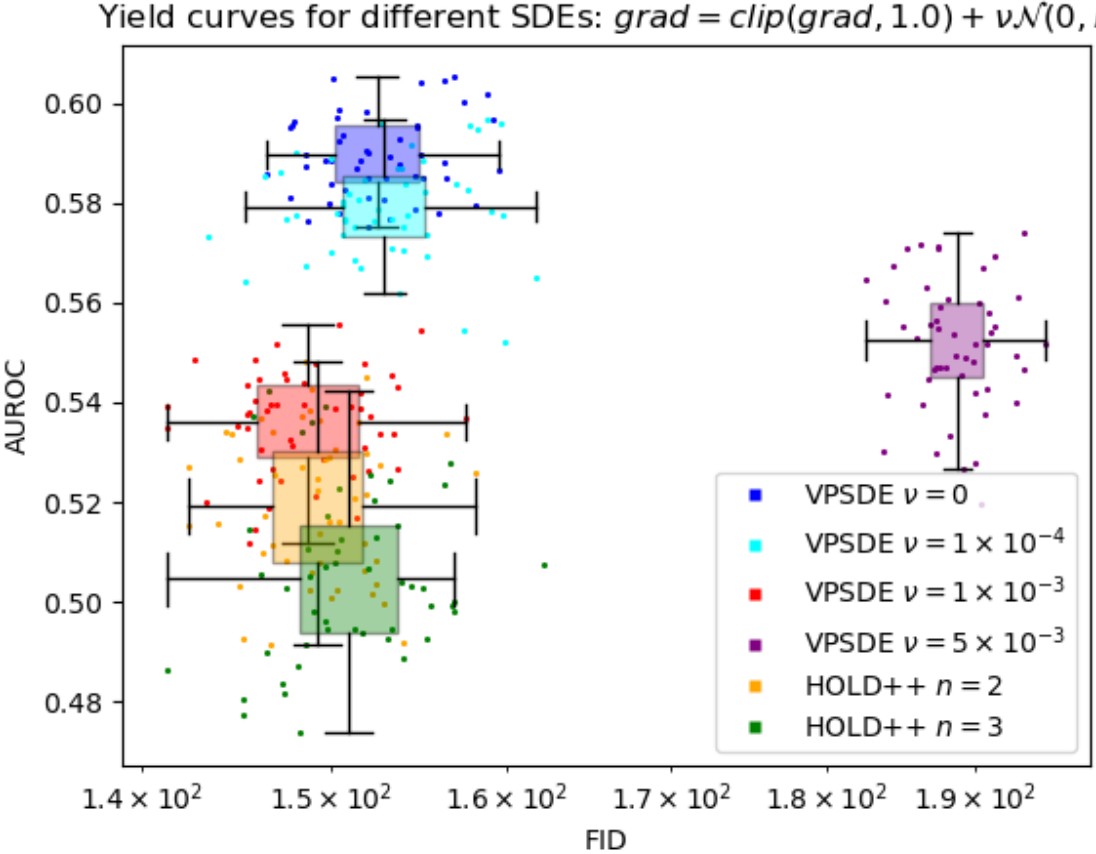

Figure 4: Yield curves on continuous diffusion models trained on 128 training images and 128 validation images of the CIFAR-10 dataset. The box and whisker plots denote the two dimensional $25^{\text{th}}$ and $75^{\text{th}}$ percentiles.

on the full CIFAR-10 dataset. Figure 3 demonstrates this on a diffusion model trained with the Variance-Preserving SDE (VPSDE). These images are generated after training for 150,000 iterations, resulting in an FID of 7.03 and AUROC of 0.503, roughly equivalent to random guessing. In these experiments, data augmentation, that is random horizontal flipping, is disabled to enhance the attacker's abilities. For the VPSDE, simple substitution yields a PIA attack metric of $R_{t,p} = \frac{\beta(t)}{2}||\mathbf{x}_t + \mathbf{s}_\theta(\mathbf{x}_t, t)||_p$. In order to validate the theory presented in this manuscript, the remaining experiments in Figure 4 are performed with 128 training images and 128 validation images taken from the CIFAR-10 dataset. The current state-of-the-art protection against MIA is the DPDM that uses DP-SGD during training to account for differential privacy. This experiment uses the DPDM strategy as a baseline, but does not use the same privacy accounting that DPDMs do because privacy in this work is analyzed on the sample level whereas privacy in the DPDM work is analyzed on the gradient level. Therefore, the baselines in these experiments are run on the VPSDE with the following operation performed on the training gradients: $\text{grad} = \text{clip}(\text{grad}, 1.0) + \nu\mathcal{N}(\mathbf{0}, \mathbf{I})$.

Figure 4 plots FIDs and AUROCs of the different models during training after they have converged to a stationary distribution. Lower FIDs indicate higher sample quality and lower AUROCs indicate lower attack performance, so the best performing models are located on the lower left-hand side of the graph. The FIDs are not on the same scale as regular state-of-the-art diffusion models because they are calculated using the 128 validation images; calculating FIDs on the full validation dataset for each point was computationally

intractable, and doing so would not add any comparative value. Figure 4 demonstrates that training models with HOLD++ results in superior performance when compared to the VPSDE trained models over the chosen $\nu$. For the sake of argumentation, if there was a parameter $\nu$ that results in a better model than HOLD++, finding such a $\nu$ is expensive, and there is no better way to find it than by training many separate models. Comparatively, HOLD++ does not require any hyperparameter tuning and evades the PIA attack by its very structure. For implementation and architecture specific details, please see the provided supplementary material.

## 6    Conclusion

It is well known that regularization helps to prevent membership inference attacks in generative models. This work provides a way to implicitly regularize using the diffusion process itself, without requiring direct data augmentation. This method works additionally well because existing membership inference attacks on diffusion models rely on the score being deterministically derived from the score network. The HOLD++ score network only models the score of the very last auxiliary variable, which means that it is not possible to run an attack with a fully deterministic score. The paper also demonstrates that this lack of deterministic score may be paired with the concept of differential privacy to help reduce membership privacy loss without a significant loss in generated data quality, making HOLD++ a practical alternative to DPDMs.

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
