# OpenReview forum: "Defending Diffusion Models Against Membership Inference Attacks via Higher-Order Langevin Dynamics"
_TMLR — Withdrawn by Authors_

### Review · Reviewer_zoSk · 2026-06-05

**Summary Of Contributions:**

**Summary**\
This paper proposes a novel approach utilizing HOLD++ to defend diffusion models against Membership Inference Attacks (MIA). The authors theoretically demonstrate that the HOLD++ framework satisfies Rényi Differential Privacy by hiding deterministic score information, and they attempt to empirically evaluate its defense capabilities against Proximal Initialization Attacks (PIA).


**Strengths**
1. **Clear narrative flow**: The general sections explaining the overarching concepts and the flow of the paper are well-structured and easy to follow.
2. **Solid theoretical foundation**: The authors provide a rigorous mathematical proof demonstrating that the HOLD++ framework satisfies Rényi Differential Privacy constraints.


**Weaknesses**
1. **Lack of detailed explanations in problem formulation**: Variables are frequently used without clear definitions, and the text assumes heavy prerequisite knowledge. Adding brief, intuitive explanations would significantly improve readability for a broader audience.
2. **Insufficient and unrealistic empirical validation**: (i) The theoretical validations in Figures 1 and 2 rely on a simple toy dataset, where setting $\beta=10$ severely degrades the spiral shape and results in a loss of sample quality. (ii) The real-world experiments on CIFAR-10 use an extremely limited subset of only 128 training and 128 validation images. The statistical significance of FID and AUROC scores derived from such a small scale is highly questionable. (iii) The authors note that training a continuous diffusion model for 150,000 iterations on the full dataset yields an AUROC of 0.503, meaning the base model is inherently robust to PIA when given sufficient data. Evaluating the defense mechanism in an artificially constrained, vulnerable environment undermines the practical relevance of the findings.
3. **Limited baseline comparisons**: While the paper notes that Differentially Private Diffusion Models (DPDMs) using DP-SGD are the current state-of-the-art defense, a direct empirical comparison against them or other diverse protection algorithms would strengthen the evaluation. Relying primarily on a simple noise-injection method applied to the VPSDE gradient as the baseline makes it somewhat difficult to fully assess the practical effectiveness of the authors' specific defensive application of the HOLD++ framework.

**Audience:**

Yes

**Audience Explanation:**

Defending diffusion models against MIA is a highly relevant and important issue in generative AI data security. Researchers working on privacy-preserving machine learning and generative modeling would be interested in understanding the theoretical applications of higher-order Langevin dynamics as a structural defense mechanism.

**Claims And Evidence:**

No

**Claims Explanation:**

While the paper presents a clear theoretical narrative, the core claims are not sufficiently supported due to flaws in the empirical validation.

**Requested Changes:**

It would be appreciated if you could explain the points suggested by weaknesses.

---

### Review · Reviewer_Bnvk · 2026-06-15

**Summary Of Contributions:**

This paper proposes a method to defend diffusion models against membership inference attacks. Overall, the paper is not clearly written and the contributions are thus hard to adequately summarize.

**Audience:**

No

**Audience Explanation:**

Due to the lack of clarity and of comparisons against baselines, I do not believe that the current manuscript would be of interest to TMLR's audience.

**Broader Impact Concerns:**

In the first paragraph of the paper, the authors discuss benefits of making diffusion models less vulnerable to membership inference attacks. While I agree there are benefits, I also believe there are disadvantages. More specifically, if a company uses data they are not legally allowed to use to train models and protect them against membership inference attacks, this line of work could potentially allow these companies to "get away with it" by making it impossible for anyone to prove the use of illegal data. This point should at the very least be acknowledged and discussed in the paper.

**Claims And Evidence:**

No

**Claims Explanation:**

It is hard to know what exactly the claims being made are in the first place. The points below all contribute to this issue:

- The authors describe a noise mechanism $f$, but provide no explanation as to how this is used. It was unclear to me if the noise mechanism is used during training of the diffusion model as a way to noise the data, or if it is used in some other way. I believe the authors use it to train, but it is completely unclear how this is done in practice. This is a key point and it should be clarified.

- The notation in the paper is not explained enough. For example, at the top of page 3, the authors define $x_0=(q_0,p_0,s_0, \dots)$ without explaining what any of the symbols mean. Without explaining this notation, the paper is very hard to follow.

- The authors present no comparisons against existing differential privacy baselines for diffusion models nor against methods to alleviate vulnerability to membership inference attacks. Without these empirical comparisons, there is not enough evidence to convince readers of the practical usefulness of the proposed procedure.

- There is no related work section: these works are only discussed in passing in the introduction. This makes it very difficult to contextualize the paper within the literature on differential privacy for diffusion models and to assess how it differs from existing approaches.

- It is hard to understand the motivation and goals of the paper. The main contribution seems to be a method to ensure differential privacy for diffusion models (section 4), yet the introduction says " the goal of this work is to enhance the defenses of diffusion models against membership inference attacks, beyond standard differential privacy".

- The explanation after the unnumbered equation after equation 2 is very hard to follow; I did not understand the meaning of $\epsilon_n$.

- It is unclear how $x_t$ is obtained when computing the $R_{t,p}$ metric, do you stochastically get $x_t$ from $x_0$ (i.e. by running the SDE forward), or do you obtain it deterministically byrunning the corresponding ODE? You also say this metric "may be interpreted as the jump-size of the process", yet the stochastic process in question has no jumps.

**Requested Changes:**

All the points I raised above.

---

### Review · Reviewer_6rvu · 2026-07-03

**Summary Of Contributions:**

## Summary

This paper studies membership inference defenses for diffusion models through the lens of critically-damped higher-order Langevin dynamics. The main idea is that the auxiliary variables introduced by HOLD++ inject additional randomness into the diffusion process and make it harder for an attacker to construct a fully deterministic PIA-style membership inference metric. The paper adapts the proximal initialization attack to the HOLD++ setting, presents an RDP-style analysis of the forward diffusion mechanism, and evaluates the approach on a Swiss Roll toy dataset and CIFAR-10. The empirical results suggest that increasing the model order n and the auxiliary variance factor $\beta$ can reduce PIA AUROC in the toy setting, and that HOLD++ may offer a favorable FID-AUROC tradeoff in a small-scale CIFAR-10 experiment.

## Strengths

The paper studies an important problem. Studying defenses beyond standard DP-SGD is a worthwhile direction, especially because DP-SGD often introduces a significant privacy-utility tradeoff.

The paper also explores an interesting connection between higher-order Langevin dynamics and privacy. The observation that auxiliary variables may interfere with deterministic PIA-style attacks is conceptually interesting, and adapting the PIA metric to the HOLD++ setting is a potentially useful starting point for further analysis.

## Weaknesses

I find some claims in the submission are not fully supported by accurate, convincing and clear evidence. Please see my explanations below. My major concerns include:
1. The paper only studies the forward process of diffusion models rather than the learned score network.
2. The proof in Lemma 4.2 is not rigorously supported.
3. Empirical experiments cannot fully support the conclusion and claims.

**Audience:**

Yes

**Audience Explanation:**

Membership inference attacks against diffusion models are an important topic in privacy and generative model security, and defenses beyond standard DP-SGD are worth studying.

**Broader Impact Concerns:**

N/A.

**Claims And Evidence:**

No

**Claims Explanation:**

The paper makes a strong claim that HOLD++ can defend diffusion models against membership inference attacks and serve as a practical alternative to DPDMs. However, I do not find the current evidence sufficiently accurate, convincing, or clearly aligned with this claim.

First, the theoretical privacy analysis appears to be formulated for the forward process
$$ f(x) = \\exp(\\mathcal{F}t)x + \\eta, $$ rather than for the training algorithm that outputs the score network $s_\\theta$. This is a major gap because the membership inference attack considered in the paper exploits the trained score network and the fact that it is optimized on training data. Therefore, proving an RDP-type property for the forward diffusion process does not by itself establish a privacy guarantee for the trained model or its released score network. The paper does not provide a clear argument connecting the proposed forward-process RDP analysis to membership privacy of the trained diffusion model.

Second, the proof of Lemma 4.2, which is central to the theoretical argument, is not convincing. The lemma claims that $\\Delta f_t$ monotonically decreases with $t$, and the proof relies on the statement that $\\mathcal{F}$ is **negative definite**. However, the HOLD++ drift matrix contains skew-symmetric coupling terms, and its symmetric part appears to be only **negative semidefinite** rather than negative definite. Moreover, the proof differentiates a maximum over data pairs without sufficient justification. As a result, the conclusion that the privacy loss is maximized at $t=0$ is not rigorously established. Since the subsequent privacy bound and interpretation rely on this monotonicity claim, this substantially weakens the theoretical support for the paper’s main claims.

Finally, the experiments do not fully compensate for these theoretical issues. The full CIFAR-10 experiment shows that the continuous PIA attack is already ineffective for a standard VPSDE model, with AUROC close to random guessing. The more favorable comparison for HOLD++ is then conducted on a very small 128-training-image/128-validation-image CIFAR-10 setting, which makes it hard to conclude that HOLD++ provides a general and practical defense against membership inference attacks. Therefore, the main claims are not yet supported by sufficiently convincing theoretical or empirical evidence.

**Requested Changes:**

Based on the three weaknesses I listed above, I would appreciate further clarification on the precise privacy object and threat model. The authors should either provide a formal argument connecting the forward-process RDP analysis to privacy of the trained score network, or substantially weaken the corresponding claims. More importantly, the proof of Lemma 4.2 should be double-checked. The current proof appears to have issues with rigor. And finally, the empirical claims should be toned down or better supported.

---

### Note · Authors · 2026-07-14

**Comment:**

Thank you everyone for your detailed feedback. Upon discussing it internally, I agree that the evidence gaps are valid and have decided to withdraw at this time.

**Withdrawal Confirmation:**

I have read and agree with the venue's withdrawal policy on behalf of myself and my co-authors.